# Mathematical Model of the Working Processes of the Gas Cap of a Piston Pump Installed in the Discharge Line

Victor Shcherba [1,*] and Irina Bulgakova [2]

1  Hydromechanics and Machines Department, Omsk State Technical University, 644050 Omsk, Russia
2  Foreign Languages Department, Omsk State Technical University, 644050 Omsk, Russia; bulgakova-i@mail.ru
*  Correspondence: scherba_v_e@list.ru or vesherba@omgtu.ru; Tel.: +7-3812-65-31-77

**Abstract:** A mathematical model of the working processes occurring in the gas cap has been developed on the basic fundamental laws of conservation of energy, mass and motion, and the equation of state, both taking into account the change in the mass of the gas due to phase transitions and the solubility of the gas in the liquid, and without taking them into account with a dividing element. In addition, there was developed a mathematical model of the liquid flow from the gas cap through a pipeline of constant cross section. It was found from the results of a numerical experiment that to reduce the feed irregularity, it is necessary to increase the length of the pipeline and the crankshaft revolutions, in addition to the known ratio of the volume of gas in the cap to the working volume of the pump; an increase in discharge pressure and an increase in the diameter of the connecting pipeline increases the feed irregularity.

**Keywords:** piston pump; gas cap; discharge pressure; connecting hydraulic lines; irregularity





## 1. Introduction

For many centuries, piston pumps have been widely used in industry and in everyday life, but one of their most important characteristics is the irregularity of pump flow. The regularity of the pump flow largely determines the regularity of the movement of the working element of the hydraulic motor, and, consequently, its reliability and efficiency.

The analysis of the scientific literature showed that most of them aimed at reducing flow, and pressure fluctuations caused by positive displacement pumps are associated with the geometric optimization of the machine itself. The researchers paid attention to the design of the inlet of axial piston pumps to ensure a smooth transition of fluid pressure between the suction and discharge phases, and vice versa [1–3]. In gear pumps, engineers have used special grooves in the side sleeves to improve machine performance and reduce pressure pulsations [4–9].

Another way to reduce intake and pressure pulsations is to use external devices in addition to the pump and hydraulic system [10–12]. These solutions can be divided into two groups: active damping systems and passive damping systems, which this article is devoted to.

Active methods involve the use of externally powered, controlled mechanisms. The method presented in [13,14] consists of high-frequency action on the angle of an oscillating plate in an axial piston pump through a switching valve, to soften the liquid intake. Various research efforts are investigating the application of piezoelectric actuators to control pistons, thereby neutralizing reverse flow by generating an anti-phase flow signal [14–16].

Their advantage is that they give effective results regardless of pressure and RPM (revolution per minute) settings, but they tend to be complex and expensive.

Passive methods are simpler and cheaper, because they do not require external control or power and do not need sensors for measurements. Mostly, passive methods dampen flow pulsations using elastic components that interact with the flow. However, they are

often designed to provide maximum performance within a certain operating range, beyond which their efficiency decreases.

Shan's study [17] demonstrates the above, and it stated where a spring accumulator is used to reduce the fundamental harmonic of the flow pulsation generated by an axial piston pump. The mechanism additionally uses the phase characteristics of suction and discharge flow pulsations, as well as pressure pulsations to improve performance. As an alternative, there was proposed a separate approach [16] in which the length and stiffness of elastic pipes are optimized in a hydraulic circuit to reduce pulsations.

This article investigates the potential of gas-filled bladder accumulators as a passive device to reduce flow and pressure pulsations. The practice of introducing accumulators into hydraulic systems to reduce noise and irregularities carried by the fluid is widespread [18,19]. By acting as a low-pass filter in a liquid-filled line, the accumulator can compensate for peaks, both negative and positive, occurring in the flow, by accumulating or intaking an excess or deficiency of liquid.

Many works devoted to the study of the operation of gas caps are performed on the calculation and analysis of the operation of air caps and they can be divided into three groups. The first group implements a mechanical approach; its fundamental work on the calculation of the gas cap is [4,20]. As the research was carried out when there were no computers and no development of the version of numerical methods of mathematical analysis, the author had to use assumptions to obtain analytical solutions. These studies were reduced to the study of the dynamics of the liquid level in the gas cap under the assumption that the water column is a material point.

It should be noted that gas caps are also used to dampen oscillatory processes in connecting hydraulic and pneumatic pipelines. These works include the study and damping of hydraulic shock [21–23]; the study and damping of oscillatory pressure processes in pipelines of reciprocating and compressor units [24–26]. An oscillatory approach is implemented in these works with consideration of resonance phenomena. In most cases, the solution of these issues was carried out using an analogy with electrical systems [27].

It should be noted that in a real gas cap there are processes of heat and mass transfer between the gas and the working fluid, and heat exchange processes between the gas and the cap surface at variable values of the heat transfer coefficient. Moreover, there is unsteady movement of a viscous fluid in the connecting pipelines, including the turbulent flow mode. To solve such a complex problem, it is necessary to use an approach based on the basic laws of thermodynamics, mass transfer, fluid and gas mechanics. This approach is the third main approach and it brings the mathematical model closer to reality and obtains the results closest to reality. It is proposed to place a gas cap directly in the pump cylinder, which will increase the efficiency of its operation while increasing the uniformity of the pump flow and reducing the weight and size indicators of the pumping unit [28]. Thus, in this paper, the task is to develop a mathematical model of the working processes of a gas cap installed on the discharge of a piston pump, based on the general laws of fluid and gas mechanics, and the theory of heat and mass transfer, to conduct a numerical experiment on its basis, in order to analyze the influence of the main geometric dimensions (relative volume of the gas cap, diameter and length of the connecting pipeline), as well as the number of revolutions and its discharge pressure on the pump flow irregularity.

## 2. Materials and Methods

In general, when developing a mathematical model of the working processes in the gas cap of a piston pump, four main objects can be distinguished: the piston pump (1), the connecting pipeline (2) between the pump and gas cap, the gas cap (3) and the connecting pipeline from the gas cap (4) (see Figure 1).

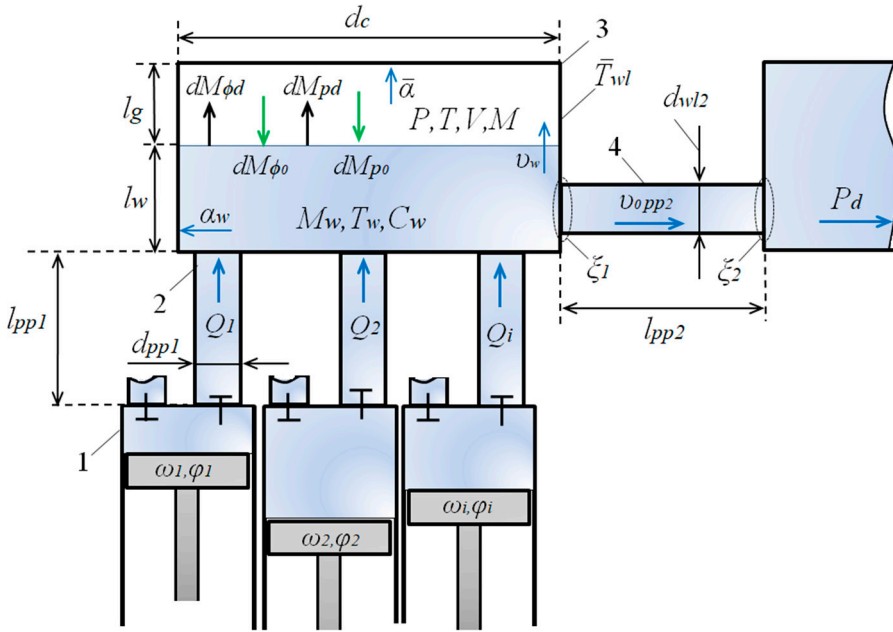

**Figure 1.** Design scheme of the gas cap of the piston pump.

The piston pump provides liquid supply from the working chamber to the gas cap through connecting pipeline 2. When performing calculations, we assume that the decrease in the volume of the delivered liquid due to leaks, underfilling of the working chambers and compressibility is negligible. Then, the liquid delivered to the cap can be determined as:

From working chamber 1

$$Q_1 = v_{n1} f_{n1} = \frac{S_{h1}}{2} \omega_1 \left( sin\varphi_1 + \frac{\lambda_1}{2} sin2\varphi_1 \right) \frac{\pi d_1^2}{4} \tag{1}$$

From working chamber 2

$$Q_2 = v_{n2} f_{n2} = \frac{S_{h2}}{2} \omega_2 \left( sin\varphi_2 + \frac{\lambda_2}{2} sin2\varphi_2 \right) \frac{\pi d_2^2}{4} \tag{2}$$

From *i*-th working chamber

$$Q_i = v_{ni} f_{ni} = \frac{S_{hi}}{2} \omega_i \left( sin\varphi_i + \frac{\lambda_i}{2} sin2\varphi_i \right) \frac{\pi d_i^2}{4} \tag{3}$$

In most practical cases, $S_{h1} = S_{h2} = S_{hi}$, $\lambda_1 = \lambda_2 = \lambda_i$, $\varphi_2 = f(\varphi_1)$; $\varphi_i = f(\varphi_1)$.

### 2.1. Mathematical Model of Working Processes in the Gas Cap

We consider the most general case, when there is no dividing element between the gas phase and the liquid phase in the gas cap.

The calculation of thermodynamic parameters in the gas phase of the air cap is similar to the calculation of the change in thermodynamic parameters in the processes of compression and expansion in a reciprocating compressor with a liquid piston.

Currently, three main approaches are used to calculate the working processes in reciprocating compressors: polytropic approximation, mathematical models with lumped parameters and mathematical models with distributed parameters [29].

The gas cap is a stage of a reciprocating compressor with a liquid piston. Mathematical models with distributed parameters determine the main thermodynamic parameters at any point in the volume of compressible gas. However, their application is limited by changing the computational grid at each moment of time, which requires time to implement this

approach and efforts to compile this program. Due to the low speed of the liquid piston compared to the speed of sound, with which pressure surges propagate, the resulting uneven distribution of thermodynamic parameters quickly disappears and, accordingly, its value tends to zero. For this reason, when calculating the working processes of reciprocating compressors, mathematical models with lumped parameters are used [29]. The developed mathematical model of working processes in a gas cap is based on the fundamental equations of conservation of energy, mass and motion, and considers many factors that are not taken into account in other models, namely: first-order phase transitions (condensation–evaporation), change in the heat transfer coefficient between gas and the walls of the gas cap in time, and change in the heat transfer coefficient between the liquid and the walls of the gas cap.

The main assumptions made are considered and their detailed analysis is given in [29]. Among the assumptions made, the following can be distinguished: the working processes are balanced and reversible, the compressible gas is homogeneous and there is no uneven distribution of temperatures and pressures in the compressible gas. The mathematical model with lumped parameters is based on the general equations of conservation of energy, mass, volume and the equation of state.

Taking into account that the kinetic energy of added and separated masses is insignificant, the energy conservation equation is transformed into the first law of thermodynamics of a body of variable mass for an open thermodynamic system in the presence of first-order phase transitions and gas solubility. Then,

$$dU = dQ - dL + i_{phad}dM_{phad} - i_{pho}dM_{pho} + i_{gr}dM_{gr} - i_o dM_{go} \tag{4}$$

The elementary external heat transfer is determined in accordance with the Newton–Rahman law, as

$$dQ = \alpha F_1\left(\overline{T}_{wl} - T\right)d\tau + \alpha F_2(T_w - T)d\tau \tag{5}$$

where $F_1 = \frac{\pi d_c^2}{4} + \pi d_c l_r$, $\left(F_2 = \frac{\pi d_c^2}{4}\right)$; $\overline{T}_{wl} = \frac{\int_{F_1} T_{wl}(F)dF}{F_1}$ is temperature averaged over the surface of the walls in the cap ($T_{wl}$ is determined experimentally [29]).

The determination of the heat transfer coefficient in the cylinders of reciprocating compressors is carried out experimentally for single-acting reciprocating compressors with a diameter of $(0.1 \div 0.22)$ m; number of revolutions $(1000 \div 1500)$ rpm, the Prilutsky–Fotin formula is widely used in the form [30]:

$$Nu(\varphi) = ARe^x(\varphi) + B \tag{6}$$

where $Nu = \frac{\alpha d_c}{\lambda}$ is Nusselt number; $Re = \frac{v_w d_c}{\mu/\rho}$ is Reynolds number.

The free surface velocity in the gas cap can be determined using the continuity equation for the liquid phase.

$$dM_w = \sum dM_{ewi} - dM_{ow} - dM_{phad} + dM_{pho} - dM_{gr} + dM_{go} \tag{7}$$

where $\sum dM_{ewi} = \rho_w \cdot d\tau \cdot \sum Q_i$ is the mass of liquid entering the gas cap from the pump cylinders.

Considering (7), $v_w$ is determined as

$$v_w = \frac{dM_w}{F_2 d\tau \cdot \rho_w} \tag{8}$$

The deformation work dL is determined as

$$dL = pdV \tag{9}$$

The elementary change in the volume of the gas phase in the cap over $d\tau$ time can be determined as

$$dV = -v_w F_2 d\tau \tag{10}$$

The mass of the gas phase in the cap: The change in the mass of the gas phase in the cap without external leaks through the leaks of the cap is due to condensation or evaporation of the working fluid (phase transitions of the first order). In this case, mass transfer is carried out by concentration diffusion, thermal and barodiffusion. Taking into account the accepted assumption of constant pressure and temperature throughout all of the gas cap, there will be no thermal nor barodiffusion.

Taking into account that the mutual motion of the phases in the gas cap is negligibly small, then to calculate the mass flows during concentration diffusion, we use Fick's first law:

$$dM_{ph} = \beta_{ph} F_2 (C_w - C_2) d\tau \tag{11}$$

Values $C_w$ and $C_2$ can be determined as

$$C_w = p_{elst} / (R_s T_w) \tag{12}$$

$$C_2 = p_s / (R_s T) \tag{13}$$

where $p_{elst}$ is steam elasticity pressure at the surface of a liquid (is a function of the temperature of the liquid and the curvature of the interface and is determined by the Clausius–Claiperon equation [31]).

The mass transfer coefficient $\beta_{ph}$ is determined based on [32]

$$\beta_{ph} = \frac{\alpha}{C_p \rho a_T / D} \tag{14}$$

It should be noted that the solubility of gas in liquid is generally described by the Henry equation and increases with increasing pressure.

Thus, taking into account the above, the equation for the conservation of the mass of the gas phase in the cap can be written:

$$dM = dM_{phad} - dM_{pho} + dM_{gr} - dM_{go} \tag{15}$$

State equation: It is well known that air follows the equation of state of an ideal gas up to 10 MPa and temperature up to 600 K. In this case, we have

$$pV = MRT \tag{16}$$

and also, taking into account that the specific internal energy and specific enthalpy for an ideal gas depend only on temperature, we have

$$dU = d(MC_v T) = C_v T dM + C_v T dM \tag{17}$$

$$i = C_p T \tag{18}$$

$$p = (k-1)\frac{U}{V} \tag{19}$$

If the pressure in the gas cap is more than 10 MPa, it is necessary to introduce the compressibility factor into the equation of state and use one of the existing equations of state for an ideal gas: van der Waals, Berthelot, Dupre, Clausius or Vukalovich–Kirillin. It must be remembered that for a real gas, $u = f(v, T)$ and $i = f(p, T)$, where $v = \frac{1}{\rho}$ is specific volume.

If there is a dividing element, there will be no mass transfer processes and the system of equations to calculate changes in thermodynamic parameters will be written as

$$\begin{cases} dU = dQ - pdV \\ dV = -(\Sigma dM_{nwi} - dM_{sw})/\rho_w \\ p = (k-1)U/V \\ T = pV/(MR) \end{cases} \tag{20}$$

Liquid phase: the pressure in the liquid phase is defined as

$$p_w = p + \Delta p \tag{21}$$

Overpressure $\Delta p$ is due to the elastic force of the dividing element:

$$\Delta p = \frac{W_{els}}{F_2} = \frac{C_{stf}(l_w - l_{w0})}{F_2} \tag{22}$$

If there is no dividing element then $p_w = p$.

To determine the temperature of the liquid in the cap, we use the energy conservation equation, if there is no dividing element.

The studies carried out prove that the pressure fluctuations do not exceed 5% in the cap [10] from the average pressure in the cap. The liquid enters the gas cap being compressed in the pump. We calculate the relative change in the volume of a liquid due to its compressibility. We assume that the nominal pressure at the outlet of the piston pump is 40 MPa. Accordingly, the increase in pressure in the gas cap will be 2 MPa. Then, the relative change in volume will be equal to $\frac{\Delta V}{V} = \frac{\Delta p}{E} = \frac{2 \cdot 10^6}{2 \cdot 10^9} = 0.1\%$, i.e., the relative change in volume is much less than one percent and can be neglected without loss of accuracy in the results.

$$dU_w = dQ_w - pdV_{wc} + \sum i_{adwi}dM_{wi} - i_{ow}dM_{0w} + i_{phad}dM_{phad} - i_{pho}dM_{pho} + i_{gr}dM_{gr} - i_o dM_{go} \tag{23}$$

Taking into account that the liquid compression in the gas cap is negligibly small, then $dV_{wc} = 0$ and the deformation work is equal to 0. The total internal energy can be determined as

$$U_w = u_w \cdot M_w \tag{24}$$

Consequently,

$$dU_w = d(u_w \cdot M_w) = M_w du_w + u_w dM_w \tag{25}$$

$du_w$ is written as

$$du_w = \left(\frac{\partial u_w}{\partial v_w}\right)_T dv_w + \left(\frac{\partial u_w}{\partial T_w}\right)_{v_w} dT_w \tag{26}$$

Considering that the fluid is incompressible and $dv_w = 0$, Equation (25) is converted to:

$$dU_w = C_w M_w dT_w + C_w T_w dM_w \tag{27}$$

where $\left(\frac{\partial u_w}{\partial T_w}\right)_{v_w} = C_w$—specific heat capacity of a liquid.

Taking into account (27), the equation for determining the change in liquid temperature can be written as:

$$dT_w = \frac{1}{C_w M_w}\left[dQ_w + \sum i_{adwi}dM_{wi} - i_{0w}dM_{0w} + i_{phad}dM_{phad} - i_{ph0}dM_{ph0} + i_{gr}dM_{gr} - i_0 dM_{p0}\right] \tag{28}$$

For a gas cap with a dividing element, Equation (28) is converted to:

$$dT_w = \frac{1}{C_w M_w}\left(dQ_w + \sum i_{adwi}dM_{wi} - i_{0w}dM_{0w}\right) \tag{29}$$

In Equations (28) and (29), the specific enthalpies are determined as

$$i_{adwi} = C_w T_{adi}; \quad i_{0w} = C_w T_w \tag{30}$$

The elementary external heat transfer $dQ_w$ is determined as

$$dQ_w = \bar{\alpha}_w F_w (\overline{T}_{wl} - T_w) d\tau \tag{31}$$

The heat transfer coefficient $\alpha_w$ for convective heat transfer depends on the flow mode for a round pipe and is defined as [33] for laminar fluid flow:

$$\bar{\alpha}_w = \frac{\alpha_w}{d_c} \left[ 0.33 Re_w^{0.3} Pr_w^{0.43} (Pr_w / Pr_{wl})^{0.25} \right] \tag{32}$$

and for turbulent flow mode:

$$\bar{\alpha}_w = \frac{\alpha_w}{d_c} \left[ 0.021 Re_w^{0.8} Pr_w^{0.43} (Pr_w / Pr_{wl})^{0.25} \right] \tag{33}$$

where $Re_w = \frac{v_w d_K}{\mu_w / \rho_w}$ is the Reynolds number; $Pr_w = \frac{\mu_w C_w}{\alpha_w}$ is the Prandtl number; and $Pr_{wl}$ is the Prandtl number at wall temperature.

The heat exchange surface with a cylindrical shape of the cap is determined as

$$F_w = l_w \pi d_c + \frac{\pi d_c^2}{2} \tag{34}$$

Mass Conservation Equation: if there is no dividing element in the cap, it is determined by Equation (7); in another case we have

$$dM_w = \sum dM_{adwi} - dM_{ow} \tag{35}$$

*2.2. Mathematical Model of Liquid Flow in the Pipeline from the Gas Cap*

Currently, various models are used to describe the flow of liquid in a pipeline, ranging from the simplest ones based on the energy conservation equation (Bernoulli), both without taking into account inertial pressure losses and taking them into account, to complex ones using two-parameter turbulence models: k-$\varepsilon$, k-$\omega$, SST and others.

When developing a mathematical model of fluid flow in a connecting pipeline, we use the principle of hierarchy and consider the calculation of the flow based on the Bernoulli equation and the unsteady one-dimensional flow of a viscous incompressible fluid.

In accordance with [4,20], the integral of the Bernoulli equation can be written as

$$\int_{l_{pp2}} \frac{\partial}{\partial l} \left( z + \frac{p}{j} + \frac{v_{pp2}^2}{2g} \right) dl + \frac{1}{y} \int_{l_{pp2}} \frac{\partial v_{pp2}}{\partial \tau} dl + \Sigma \Delta h_i = 0 \tag{36}$$

where $\Sigma \Delta h_i = \left( \Sigma \xi_i + \lambda_{pp2} \frac{l_{pp2}}{d_{pp2}} \right) \frac{v_{pp2}^2}{2g}$—head losses due to local and hydraulic resistance along the length.

The solution of this equation was carried out in [34].

Without taking into account the forces of inertia, the equation for determining the velocity of the liquid in the pipeline is written as:

$$v_{pp2} = \sqrt{\frac{2g \left[ \left( z_{1pp2} + \frac{p}{\rho_w g} \right) - \left( z_{2pp2} + \frac{P_d}{\rho_w g} \right) \right]}{\left( \lambda_{pp2} \frac{l_{pp2}}{d_{pp2}} + \Sigma \xi_i \right)}} \tag{37}$$

The coefficient of friction along the length $\lambda_{pp2}$ is a function of the Reynolds number [35] and, accordingly, $v_{pp2}$. As a result, Equation (37) must be solved at each time step by the successive approximations method.

The system of differential equations for describing an unsteady one-dimensional flow of a viscous incompressible liquid can be written as a system of equations of motion and continuity [36–38]:

$$\rho_w \frac{\partial Q_{wpp2}}{\partial \tau} + f_{pp2} \frac{\partial p}{\partial x} + \frac{\lambda_{pp2} \rho_w}{2 d_{pp} f_{pp2}} Q_{wpp\text{TP2}} |Q_{w\text{TP2}}| = 0 \tag{38}$$

$$\frac{\rho_w a^2}{f_{pp2}} \frac{\partial p}{\partial x} + \frac{\partial p}{\partial \tau} = 0 \tag{39}$$

We recommend to solve this system by the method of "characteristics".

Boundary conditions at the ends of the pipeline are adjacent, on the one hand, to the gas cap, and on the other side to the consumer of the liquid in the form of pressures: in the gas cap—$p_w$; at the consumer of the liquid—$p_d$.

Having defined $v_{pp2}$ or $Q_{wpp2}$, the value of $dM_{ow}$ is determined as

$$dM_{ow} = \rho_w v_{pp2} f_{pp2} d\tau = Q_{wpp2} \rho_w d\tau \tag{40}$$

*2.3. Verification of the Developed Model*

Currently, there are two research methods: experimental and theoretical. Both research methods have an error in determining the true results. Until now, the experimental method is believed to correspond to reality most accurately, and the developed models have been verified on the results of experimental studies. In general, we also consider such an approach proper if a comprehensive assessment of the error in the determination of results in the experiment has been carried out. In some cases, the error in experimental studies exceeds the error in the mathematical models based on the known laws of conservation of energy, mass and motion. If experimental verification is not available, then the developed mathematical mode can be verified by the following methods:

1  Compare it with the results of experimental studies of other authors.
2  Compare it with the results of theoretical studies of other authors.

Both approaches are not available in this work. There are no complete results of experimental studies on the influence of the main design and operational parameters of the gas cap in the literature, and those that exist are theoretical and experimental results that correspond to the results obtained in this article (see Bashta, T. M. Displacement pumps and hydraulic engines of hydraulic systems: textbook for universities/T. M. Bashta.—Moscow: Mashinostroenie, 1974.—606 p.) [10]. When using the results obtained on other mathematical models, there is a question of how much the existing developed mathematical models are closer to real results than the data. In our opinion, they are further from reality, because mainly they use the theory of oscillatory processes, which do not consider many physical phenomena (heat and mass transfer, etc.).

## 3. Results and Discussion

We conducted a numerical experiment to analyze the influence of the main operational and design parameters on the irregular pump flow and the ongoing work processes in the gas cap. As a study object we took a single-acting piston pump with the following main dimensions: piston stroke—0.1 m; piston diameter—0.1 m; the ratio of the piston stroke to twice the length of the connecting rod—0.2. The piston pump was connected to a gas cap by a pipeline with a diameter of 0.03 m and a length of 0.5 m. The gas cap had a dividing element with an infinitesimal rigidity. We chose water and air as the working fluid. The gas cap was made in the form of a cylinder 0.6 m long. When the volume of the gas cap was changed, its diameter was changed. The initial height of the liquid level was chosen

equal to 1/3 of the total height of the tank. To simplify the calculations and due to the smallness of its change, the temperature of the liquid in the cap was assumed to be 293 K, and the average surface temperature of the cap walls was assumed to be 300 K. The gas cap helped to maintain a constant pressure over time ($p_d$) by a pipeline, the inner diameter and length of which varied, and the value of the roughness of the inner walls remained constant and equal to 0.000005 m. The preliminary analysis of the impact on the working processes in the cap and the irregular pump flow established that the crankshaft revolutions of the pump and the discharge pressure can be taken as independent operational parameters. We chose as the main geometric parameters: the volume of the gas cap due to a change in its diameter, and the diameter and length of the connecting pipeline. Thus, we obtained five independent variables: $p_d$, $n_{rev}$, $d_{pp2}$, $l_{pp2}$, $V_g/V_h$.

$V_g/V_h$ is the ratio of the initial gas volume to the working volume $V_h$ of the pump.

As response functions, we chose: feed irregularity ($\Delta = \frac{Q_{max}-Q_{min}}{Q_{av}}$), maximum fluid flow ($Q_{max}$), minimum flow rate ($Q_{min}$), the average pressure in the cap ($p_{av} = \frac{\int_0^{2\pi} P(\varphi)d\varphi}{2\pi}$) and the average gas temperature in the cap ($T_{av} = \frac{\int_0^{2\pi} T(\varphi)d\varphi}{2\pi}$).

The classical plan with fractional replicas was chosen for the numerical experiment.

### 3.1. Features of the Implementation of the Mathematical Model

In general, the developed mathematical model is a set of first-order differential equations of total derivatives. This system does not have an analytical solution and well-known numerical methods can be used to solve it; for example, Euler, Runge–Kutta (of different orders of accuracy), Adamson, Monte Carlo, etc. In this article, the Euler method was used when solving differential equations.

As a base point, we take a point with the following independent variables: $V_g/V_h$ = 64; $l_{pp2}$ = 2 m; $d_{pp2}$ = 0.03 m; $n_{rev}$ = 300 rpm. The calculation starts from the crankshaft rotation angle $\varphi = \pi$; in this case, the pressure and temperature in the gas cap were set arbitrarily, while the pressure in the gas cap was set not much higher than the discharge pressure ($p_d$), and the temperature was set equal to the temperature of the liquid. The instantaneous supply of liquid from the pump cylinder through pipeline 2 to gas cap 3 was calculated. Then the gas cap and pipeline 4 were calculated. When calculating the pipeline, an equation was used to determine the fluid flow rate without taking into account inertial losses. The calculation was carried out with a step according to the crankshaft rotation angle $\Delta\varphi = 2\pi/7200$. After the calculation for one rotation of the crankshaft, the initial and final pressure values in the gas cap were compared. If the difference in the obtained pressures was less than 1 kPa, the calculation was stopped; if not, the calculation was started anew, while the obtained final thermodynamic parameters of the gas and liquid in the cap became the initial ones. In most of the calculation options performed, the number of iterations did not exceed 25.

When determining the fluid flow rate in pipeline 4, the number of iterations was limited to 50, while the discrepancy in determining the rate was set to less than 1%. During the calculation, at each iteration, the value of the maximum flow rate in pipeline 4 during the crankshaft rotation from 0 to $2\pi$ was determined—$Q_{max}$, the minimum value of the liquid flow rate—$Q_{min}$, and the average integral value of the liquid flow rate—$Q_{av}$.

### 3.2. Analysis of the Influence of the Ratio of the Initial Volume of the Gas Phase to the Working Volume of the Pump ($V_g/V_h$)

In accordance with the previously obtained results in [1,4], an increase in the ratio $V_g/V_h$ leads to a decrease in the irregularity of the pump flow (see Figure 2). The resulting dependence is hyperbolic in nature. From the presented results, we can conclude that an acceptable feed irregularity (less than 10%) is achieved at $V_g/V_h$ more than 70%.

The dependences of the minimum and maximum values of the liquid flow in pipeline 4 presented in Figure 3 established that these dependences are parabolic in nature and

their change decreases significantly, starting from $V_g/V_h > 50$. Increasing $V_g/V_h$ leads to a decrease in the average pressure ($p_{av}$) and average gas temperature in the cap.

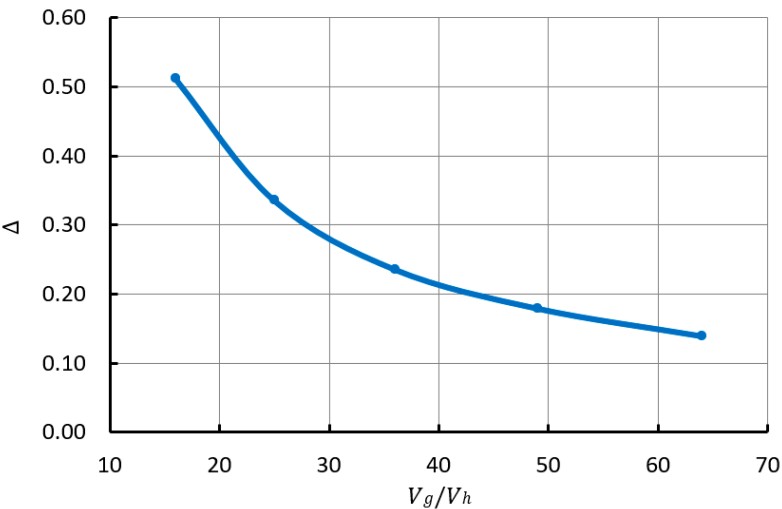

**Figure 2.** The dependence of the irregularity of the liquid supply on the ratio of the initial value of the volume of the gas phase to the working volume.

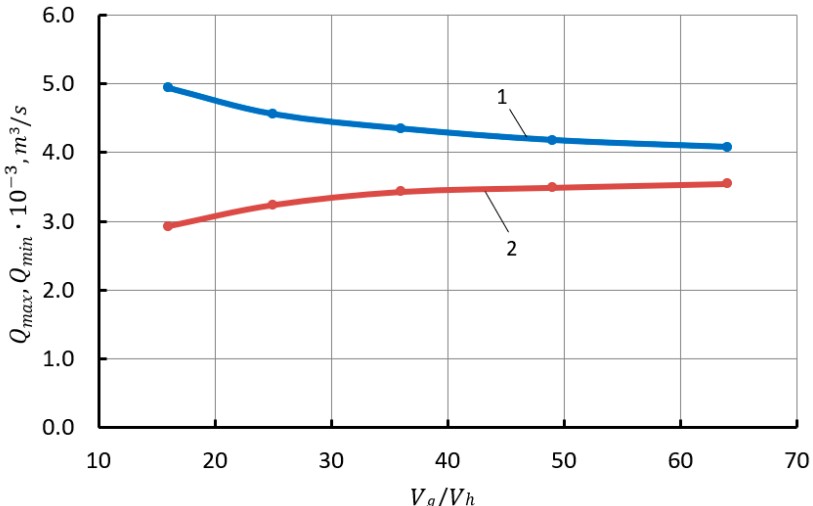

**Figure 3.** Dependence of the maximum and minimum liquid flow rate from the gas cap on the ratio of the initial value of the gas phase volume to the working volume (1—$Q_{max}$; 2—$Q_{min}$).

Figure 4 presents $\Delta p = p_{av} - p_d$ and $T_{av}$ depending on the ratio $V_g/V_h$. The obtained dependences are close to linear. Values $p_{av}(\Delta p)$ and $T_{av}$ change insignificantly and are about 5 kPa and about 0.6 K with the increasing $V_g/V_h$ from 16 to 64.

### 3.3. Analysis of the Influence of the Crankshaft Revolutions

Increasing the crankshaft revolutions increases the amount of liquid entering the gas cap per unit time. This leads to a decrease in feed irregularity (see Figure 5). The presented dependence has a clearly non-linear nature, close to parabolic. At a speed greater than 350 rpm, $\Delta$ becomes less than 10%. With the increase in $n_{rev}$ values, $Q_{max}$ and $Q_{min}$ increase almost linearly, and the difference between them decreases. As the number of revolutions increases, $p_{av}(\Delta p)$ and $T_{av}$ increase almost linearly (see Figure 6). Increasing $\Delta p$ with an increase in the number of revolutions from 200 rpm to 400 rpm is 70 kPa, the value of $T_{av}$ is even more significant and is about 5 K. An increase in $T_{av}$ is due to an increase in the frequency of fluctuations in the liquid level in the gas cap. Thus, increasing $n_{rev}$ leads to a significant increase in $p_{av}$ and $T_{av}$ with a decrease in feed irregularity.

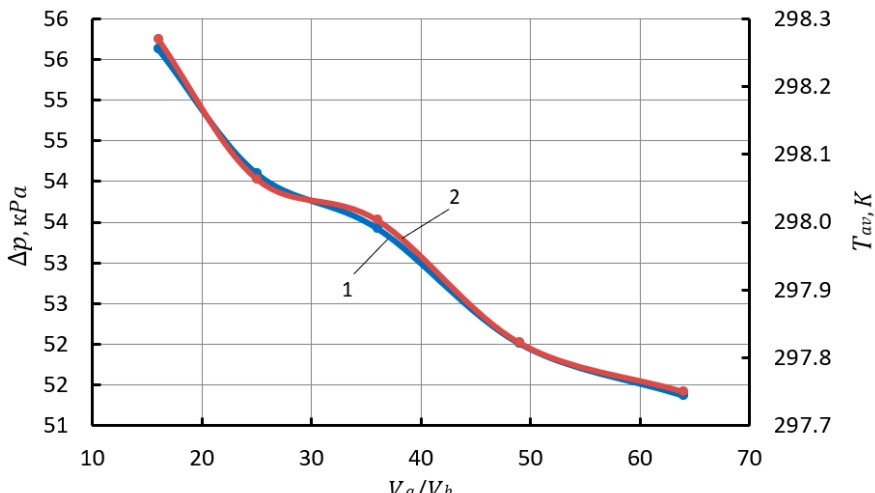

**Figure 4.** Dependence of the difference between the average pressure and the pressure in the cap ($\Delta p$), the average temperature of the gas in the cap from the ratio of the initial value of the volume of the gas phase to the working volume (1—$\Delta p$; 2—$T_{av}$).

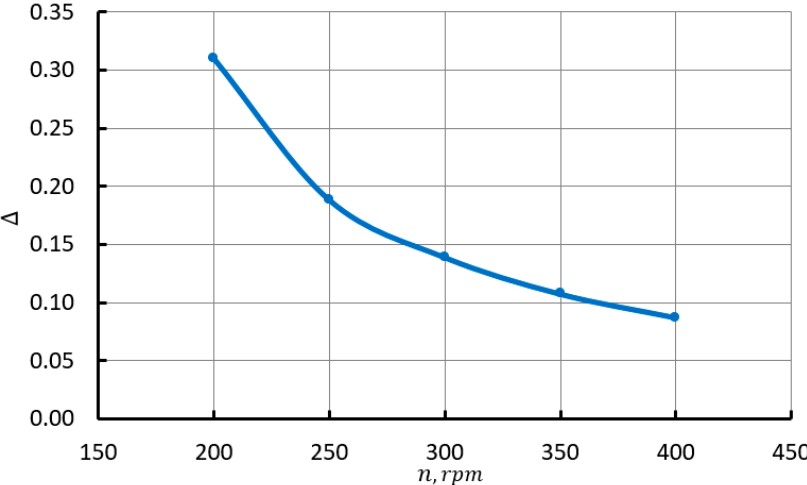

**Figure 5.** The dependence of the irregularity of liquid supply on the crankshaft revolutions.

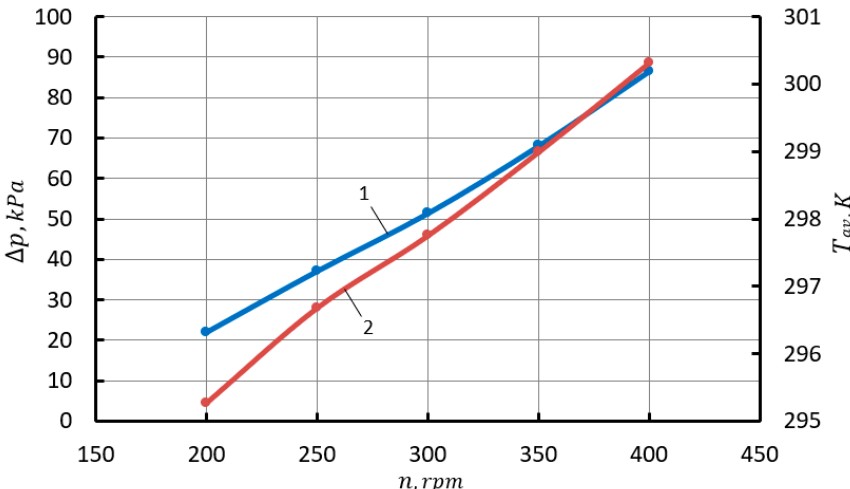

**Figure 6.** Dependence of the difference between the average pressure and the pressure in the cap ($\Delta p$), average gas temperature in the cap from the crankshaft revolutions (1—$\Delta p$; 2—$T_{av}$).

### 3.4. Discharge Pressure Analysis

With an increase in discharge pressure, the feed irregularity increases linearly (see Figure 7). This dependence is very significant. So, with an increase in discharge pressure from 1 MPa to 3 MPa, an increase in the feed irregularity occurs from 13.9% to 36.57%, i.e., also almost by 3 times.

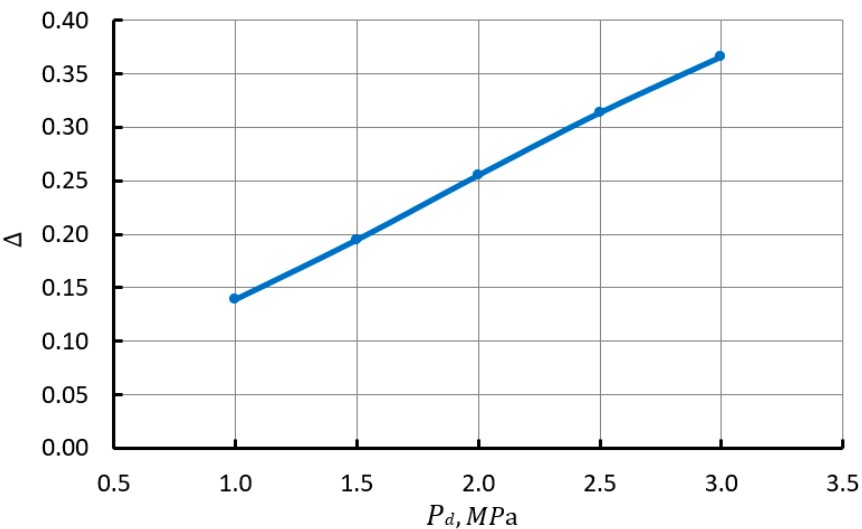

**Figure 7.** The dependence of the feed irregularity on the discharge pressure.

An increase in pressure in the gas cap is equivalent to a decrease in the volume of gas in it, and this, in turn, leads to an increase in the feed irregularity.

With an increase in discharge pressure, $Q_{max}$ and $Q_{min}$ also change almost linearly: $Q_{max}$ increases, but $Q_{min}$ decreases. With an increase in discharge pressure, $\Delta p$ increases very slightly from 51.3 kPa to 54.83 kPa. The average temperature $T_{av}$ in a gas cap with increasing $p_d$ decreases. This is due to an increase in the gas density, which leads to an increase in the Re number and an increase in the heat transfer coefficient.

### 3.5. Analysis of the Influence of the Geometric Dimensions of Connecting Pipeline 4

Increasing the diameter of the pipeline increases the liquid flow rate from the gas cap, which increases pressure fluctuations in the cap and fluid flow fluctuations from the gas cap. $Q_{max}$ with an increase in $d_{pp2}$ increases parabolically and $Q_{min}$ also decreases parabolically (see Figure 8). There is also a parabolic increase in feed irregularity with the $d_{pp2}$ increase. It should be noted that the size of the pipeline diameter has a very significant effect on the feed irregularity (see Figure 9). An increase in the diameter of the pipeline leads to a decrease in the average pressure in the cap and the average temperature. We observe a significant decrease $p_{av}$ by almost 100 kPa and temperature by 7.5 (see Figure 10).

An increase in the length of pipeline 4 leads to a decrease in the feed irregularity (see Figure 11). An increase in the length of the pipeline increases the hydraulic resistance, which leads to a decrease in the fluid velocity and its flow rate. Thus, an increase in the length of a pipeline is similar to a decrease in its diameter. The presented results allow us to conclude that the increase in length does not have a significant effect on the feed irregularity.

The decrease in feed irregularity is due to a decrease in $Q_{max}$ and increase in $Q_{min}$ (see Figure 12). An increase in the length of the pipeline leads to a linear increase in the average pressure $p_{av}$ and temperature $T_{av}$ (see Figure 13). It should be noted that the increase in $p_{av}$ and $T_{av}$ is insignificant.

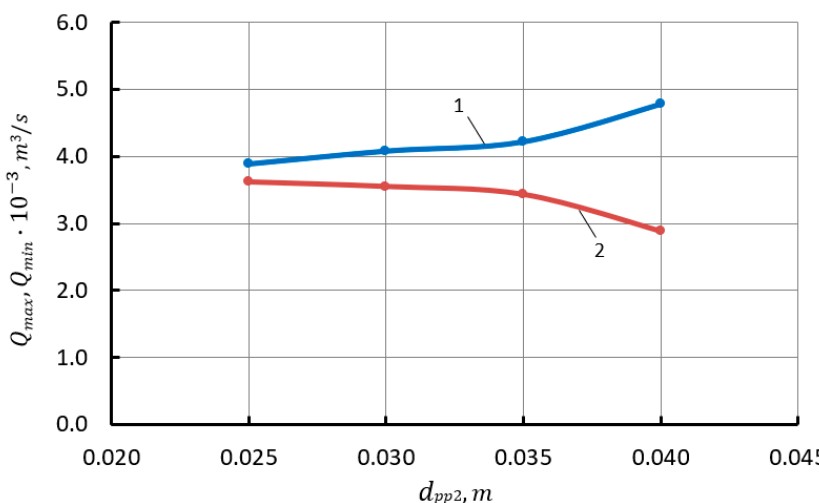

**Figure 8.** Dependence of the maximum and minimum liquid flow rate from the gas cap on the diameter of connecting pipeline 4 (1—$Q_{max}$; 2—$Q_{min}$).

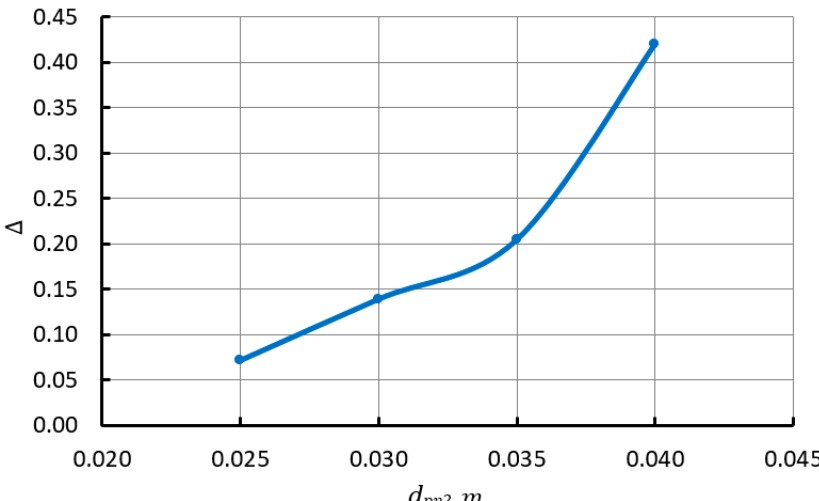

**Figure 9.** The dependence of the feed irregularity on the diameter of connecting pipeline 4.

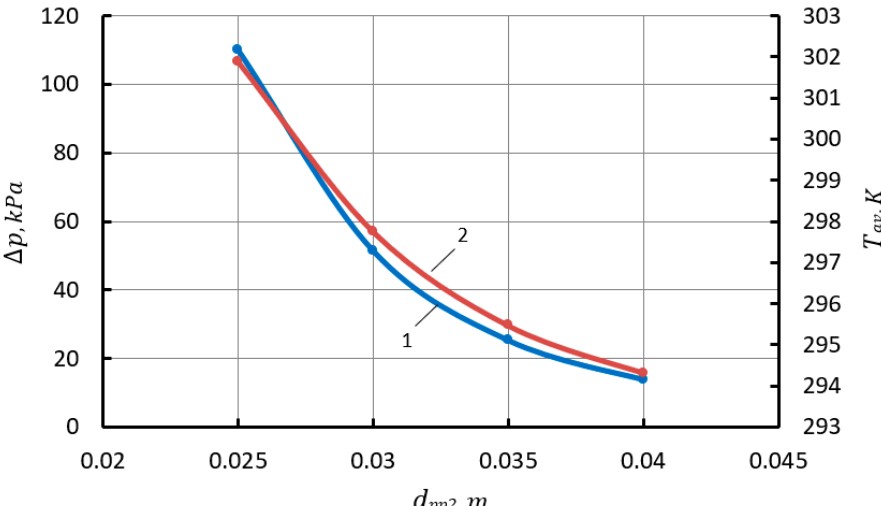

**Figure 10.** Dependence of the difference between the average pressure and the pressure in the cap ($\Delta p$), average gas temperature in the cap from the diameter of connecting pipeline 4 (1—$\Delta p$; 2—$T_{av}$).

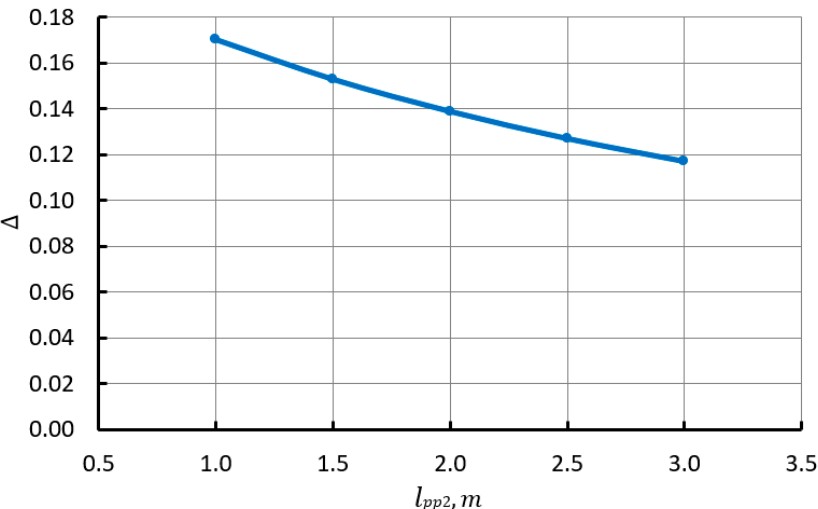

**Figure 11.** The dependence of the feed irregularity on the length of connecting pipeline 4.

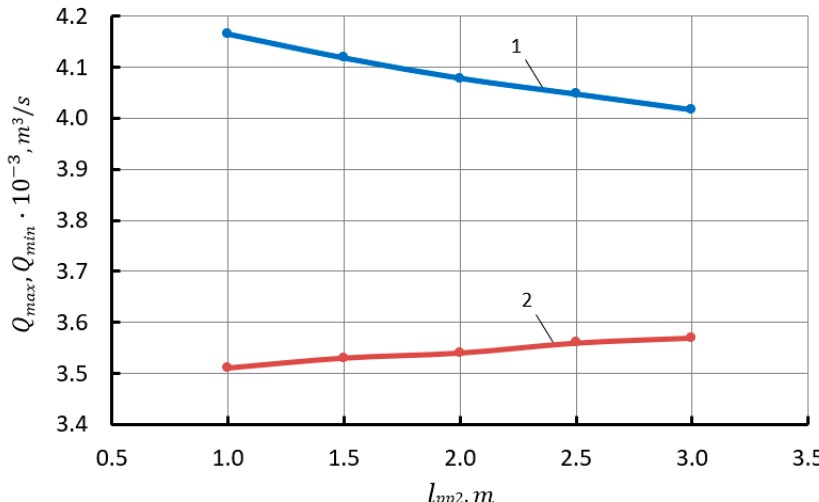

**Figure 12.** Dependence of the maximum and minimum liquid flow rate from the gas cap on the length of connecting pipeline 4 (1—$Q_{max}$; 2—$Q_{min}$).

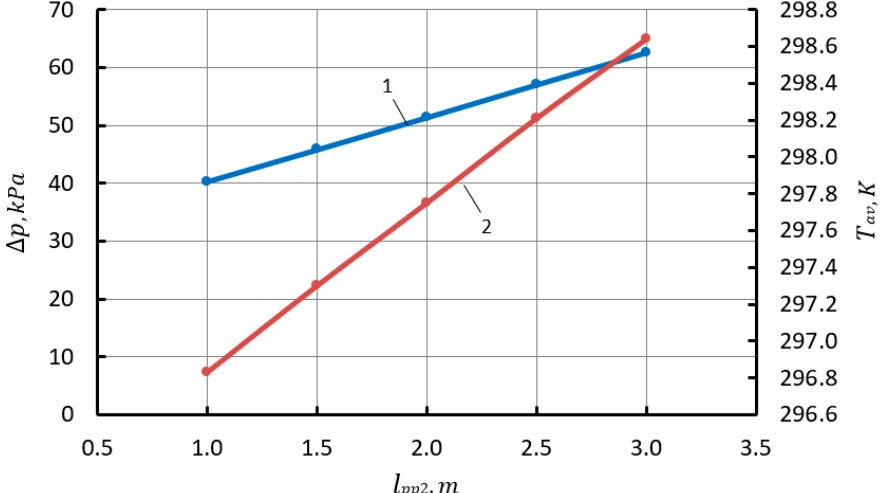

**Figure 13.** Dependence of the difference between the average pressure and the pressure in the cap ($\Delta p$), average gas temperature in the cap from the length of connecting pipeline 4 (1—$\Delta p$; 2—$T_{av}$).

### 3.6. Ranking the Influence of Independent Variables on Feed Irregularity

It is important for constructors and designers of positive displacement pumps with gas caps to know which independent variables to pay attention to in order to obtain the minimum non-uniformity of the pumping unit. As a result, it is necessary to rank the influence of independent variables on the feed irregularity of the pumping unit.

The ranking will be carried out in relation to the relative feed irregularity to the relative independent argument $\frac{\Delta\bar{\delta}}{\Delta\bar{x}}$, where $\Delta\bar{\delta} = \frac{\delta_{max}-\delta_{min}}{\delta_{av}}$; $\Delta\bar{x} = \frac{x_{max}-x_{min}}{x_{av}}$. The results of the ranking are presented in Table 1.

**Table 1.** The results of the ranking.

| x | $d_{pp}$ | $V_g/V_h$ | $n_{rev}$ | $p_d$ | $l_{pp}$ |
|---|---|---|---|---|---|
| $\Delta\bar{\delta}/\Delta\bar{x}$ | 3.073 | 0.955 | 0.94 | 0.899 | 0.371 |

The presented results allow us to draw the following conclusions:

- The diameter of pipeline 4 has the greatest influence on the feed irregularity;
- The relative initial gas volume $V_g/V_h$ is only the second value, although it is considered as the main in existing works;
- The crankshaft revolutions and the discharge pressure have approximately the same effect;
- Pipeline 4 length has the least effect;
- Thus, it is important for developers, designers and those involved in the operation of pumping units with gas caps to determine the diameter of the supply pipe (this recommendation is not available in the existing literature), and to determine the ratio of the initial gas volume in the cap to the working volume of the pump chamber (this recommendation is in the technical literature).

### 4. Conclusions

1.  We developed a mathematical model based on a thermodynamic approach which allowed detailed consideration of work processes occurring in the gas cap on the basic fundamental laws of conservation of energy, mass and motion, and the equation of state, both taking into account the change in the mass of the gas due to phase transitions and the solubility of the gas in the liquid, and without taking them into account (if there is a separating element). To close the developed mathematical model of working processes in the gas cap and to determine the maximum and minimum fluid flows from the pump, a mathematical model of the fluid flow from the gas cap through a pipeline of constant cross section has been developed;

2.  It was found on the numerical experiment that to reduce the feed irregularity, it is necessary to increase the length of the pipeline and the crankshaft revolutions, in addition to the known ratio of the initial gas volume in the cap to the pump displacement; an increase in discharge pressure and an increase in the diameter of the connecting pipeline increases the feed irregularity;

3.  The ranking of the influence of the analyzed independent variables on the feed irregularity proved that the diameter of the connecting pipeline has the greatest influence, then the ratio of the initial volume of gas in the cap to the volume of the working chamber of the pump. Regarding operating parameters, the crankshaft revolutions of the pump and the discharge pressure have approximately the same influence, and the length of the connecting pipeline has the least influence. The crankshaft revolutions of the pump and the discharge pressure of the pump have approximately the same effect on the feed irregularity of the pump. The length of the connecting pipeline between the gas cap has the least impact.

**Author Contributions:** Conceptualization, V.S.; methodology, V.S. and I.B.; software, V.S.; validation, V.S. and I.B.; formal analysis, V.S.; investigation, V.S.; data curation, V.S.; writing—original draft preparation, V.S. and I.B.; writing—review and editing, V.S. and I.B.; visualization, V.S. and I.B.; project administration, V.S. and I.B.; funding acquisition, V.S. and I.B. All authors have read and agreed to the published version of the manuscript.

**Funding:** The research was supported by the Russian Science Foundation Grant No. 22-29-00399, https://rscf.ru/project/22-29-00399/ (accessed on 10 June 2023).

**Informed Consent Statement:** Not applicable.

**Data Availability Statement:** The results of the work are new, some of the known results are in the available sources, which are cited in the article.

**Conflicts of Interest:** The authors declare no conflict of interest.

## Nomenclature

The nomenclature of the paper is shown below:

| | |
|---|---|
| $v_{n1}, v_{n2}, v_{ni}$ | 1, 2 and i-th piston speed |
| $f_{n1}, f_{n2}, f_{ni}$ | areas of these pistons |
| $\omega_1, \omega_2, \omega_i$ | angular velocities |
| $S_{h1}, S_{h2}, S_{hi}$ | full piston strokes |
| $d_1, d_2, d_i$ | piston diameters |
| $\varphi_1, \varphi_2, \varphi_i$ | angles of crankshafts rotation |
| $\lambda_1, \lambda_2, \lambda_i$ | ratio of piston strokes to crank lengths |
| dU | elementary change in the total internal energy of the gas phase in the cap |
| dQ | elementary external heat exchange |
| dL | elementary contour work |
| $i_{phad}$ | specific enthalpy of the added steam due to liquid evaporation |
| $i_{pho}$ | specific enthalpy of the separated steam during condensation |
| $dM_{phad}$ | elementary mass added to the gas phase of the cap due to first-order phase transitions |
| $dM_{pho}$ | elementary mass separated from the gas phase of the cap due to phase transitions of the first order |
| $i_{gr}$ | specific enthalpy of a gas released from a liquid |
| $i_o$ | specific enthalpy of a gas dissolved in a liquid |
| $dM_{gr}$ | elementary mass of gas released from a liquid |
| $dM_{Po}$ | elementary mass of gas dissolved in a liquid |
| F1 | the heat exchange surface of the liquid |
| $d_c$ | cap diameter |
| $l_r$ | length of the generatrix of the cylinder of the gas cap in contact with the gas |
| $F_2$ | free surface area of a liquid |
| $T_w$ | liquid temperature |
| $\lambda$ | gas thermal conductivity coefficient |
| A, x, B | constant coefficients (A = 0.2 ÷ 0.235, x = 0.8 ÷ 0.86, B = 500 ÷ 800) |
| $\mu$ | dynamic viscosity coefficient |
| $\rho_w$ | liquid density |
| $d\tau$ | elementary time |
| $dM_{ow}$ | mass of liquid separated over time $d\tau$ from the gas cap |
| $C_w$ | steam concentration on the liquid surface |
| $C_2$ | average steam concentration in the gas phase |
| $\beta_{pg}$ | mass transfer coefficient |
| $R_s$ | steam gas constant |
| $p_s$ | partial steam pressure |
| $a_T$ | thermal diffusivity |
| $C_P$ | specific isobaric heat capacity of gas |
| D | diffusion coefficient (is a function of pressure and temperature) |
| $C_{stf}$ | dividing element stiffness |
| $l_w$ | current position of the liquid level |

| | |
|---|---|
| $l_{w0}$ | the initial position of the liquid level (in most practical cases, the liquid occupies 1/3 of the cap) |
| $l_{pp2}$ | pipeline 4 length |
| $z$ | center of gravity coordinate |
| $j = \rho_w g$ | liquid specific gravity |
| $g$ | acceleration of gravity |
| $\lambda_{pp2}$ | coefficient of hydraulic friction along the length |
| $d_{pp2}$ | pipeline 4 inner diameter |
| $\Sigma \xi_i$ | sum of local coefficients (sudden expansion, sudden contraction, flow turn, etc.) |
| $z_{1pp2}, z_{2pp2}$ | centers of gravity of control sections |
| $p_d$ | fluid pressure |
| $Q_{wpp2}$ | volume flow of liquid in pipeline 4 |
| $f_{pp2}$ | cross-sectional area of the connecting pipeline |
| $a$ | sound speed |

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
