# Peer review of "Mathematical Model of the Working Processes of the Gas Cap of a Piston Pump Installed in the Discharge Line"

_inventions, doi:10.3390/inventions8040095_

Round 1

Reviewer 1 Report

Figure 1 should be supplemented with elements with parameters included in Eqs.1-3, e.g. angular velocities and angles of crankshaft rotation.

In Figure 1 Qi is the flow rate in connecting pipeline 2, and in Eq.1-3 Qi is the flow rate in working chamber of the cylinder.

The same designations are used for Q, which are flow rate and heat exchange.

The mathematical solutions are not explained.

Results and the discussion are based primarily on flow rate irregularity, but there is no such solution from the mathematical model.

Line 218: should be fluid flow rate irregularity DQ, and as defined are Qmax, Qmin and Qav.

Eq.22. What is Fels and where is the dividing element. Collision in the F notation should be either force or area once.

Chapter 3.6. should be extended.

Line 329: undefined parameters.

Nomenclature should be in alphabetical order

The conclusions are incomprehensible.

The weak side of the work is only analytical solutions and static results.

The models are based on thermodynamic processes, there is no such reference in the results.

Author Response

Response to reviewers:

Dear reviewers and editor, thank you for reviewing this manuscript. Your comments have been very helpful and have improved this manuscript. Please find below the details on how we addressed them. You may also refer to the highlighted and tracked version of the revised manuscript.

Comments of Reviewer 1

Replies

1. Figure 1 should be supplemented with elements with parameters included in Eqs.1-3, e.g. angular velocities and angles of crankshaft rotation.

We corrected Figure, angular velocities and angles of the crankshaft rotation have been added to it.

2. In Figure 1 Qi is the flow rate in connecting pipeline 2, and in Eq.1-3 Qi is the flow rate in working chamber of the cylinder.

The flow rate in connecting pipeline 2 is the same as the flow rate from the working chamber of the cylinder, because we assume that there are no fluid leaks.

3. The same designations are used for Q, which are flow rate and heat exchange.

The liquid flow rate and the elementary amount of heat are denoted by the same Q in the technical literature. It is inappropriate to accept any new designations in the article. It is quite clear from the text where Q is used to denote flow, and where it is used to denote heat transfer.

4. The mathematical solutions are not explained.

We made the corrections in the article

5. Results and the discussion are based primarily on flow rate irregularity, but there is no such solution from the mathematical model.

We made the corrections in the article

6. Line 218: should be fluid flow rate irregularity DQ, and as defined are QmaxQmin and Qav.

We made the corrections in the article

7. Eq.22. What is Fels and where is the dividing element. Collision in the F notation should be either force or area once.

We made the corrections in the article

8. Chapter 3.6. should be extended.

We made the corrections in the article

9. Line 329: undefined parameters.

We made the corrections in the article

10. Nomenclature should be in alphabetical order

We made the corrections in the Nomenclature

11. The conclusions are incomprehensible. The weak side of the work is only analytical solutions and static results.

The models are based on thermodynamic processes, there is no such reference in the results.

We made the corrections in the article

Reviewer 2 Report

The article deals with a well-known problem. The mathematical description is also not novel. No simplifying assumptions are given in the construction of the mathematical model, hence it is difficult to assess the extent to which the model results are applicable. No experimental verification. The numerical experiment is an abuse in the world of science. List of references not actual. Some next comments in attached file.

Author Response

Response to reviewers:

Dear reviewers and editor, thank you for reviewing this manuscript. Your comments have been very helpful and have improved this manuscript. Please find below the details on how we addressed them. You may also refer to the highlighted and tracked version of the revised manuscript.

Comments of Reviewer 2

Replies

1. The article deals with a well-known problem.

The issue of feed irregularity appeared along with the creation of positive displacement pumps and still remains relevant. The gas cap is one of the radical means of its reducing. Considering the above said, as well as that recently there have been few studies on the analysis of the influence and optimization of geometric and operational parameters, the relevance of this work is doubtless. 

2. The mathematical description is also not novel.

The developed mathematical model is based on the fundamental equations of conservation of energy, mass, motion, which have been known for more than one hundred years, but they continue to be used in all modern research.

3. No simplifying assumptions are given in the construction of the mathematical model, hence it is difficult to assess the extent to which the model results are applicable.

In the article, when considering the working processes in the gas cap, there is a reference with the similar assumptions discussed, and the main accepted assumptions are listed. When considering the liquid flow in connecting pipeline 4, we indicated that the liquid is viscous and incompressible.

4. No experimental verification.

Due to the basic fundamental equations and assumptions that are typical for solving similar thermodynamic problems, experimental research is relevant, but not necessary.

5. List of references not actual.

We updated References and reconsidered Introduction.

6. Links and sources should be discussed separately, because. artificial increase in the number of sources.

The authors do not increase the number of references and do not engage in self-citation. The total number of references meets the journal requirements, and a separate discussion of each of the cited sources will increase the volume of the article without obtaining new information.

7. Formula 1 - And what about compressibility?

We do not consider the compressibility of the liquid in the article.

Round 2

Reviewer 1 Report

The authors took into account all the comments and suggestions contained in the review.

The manuscript may be published in its present form.

Author Response

Dear reviewer, thank you for your comments and the work done.

Reviewer 2 Report

Thank you for your responses to my comments. I find the replies unsatisfactory. In my opinion, the concept of the paper should be redesigned.

Author Response

Dear reviewer, thank you for your comments and the work done. In the file we have given answers to your comments.
